# Outcomes and Reliability of Perforator Flaps in the Reconstruction of Hidradenitis Suppurativa Defects: A Systemic Review and Meta-Analysis

**DOI:** 10.3390/jcm11195813

**Published:** 2022-09-30

**Authors:** Camille Vaillant, Yanis Berkane, Elise Lupon, Michael Atlan, Pascal Rousseau, Alexandre G. Lellouch, Jérôme Duisit, Nicolas Bertheuil

**Affiliations:** 1Department of Plastic, Reconstructive and Aesthetic Surgery, CHU Rennes, University of Rennes 1, 35000 Rennes, France; 2Department of Plastic, Reconstructive and Aesthetic Surgery, CHU Angers, 49933 Angers, France; 3Vascularized Composite Allotransplantation Laboratory, Massachusetts General Hospital and Shriners Children’s Boston, Harvard Medical School, Boston, MA 02115, USA; 4Department of Plastic, Reconstructive and Aesthetic Surgery, Rangueil Hospital, CHU Toulouse, 31013 Toulouse, France; 5Department of Plastic, Reconstructive and Aesthetic Surgery, Tenon Hospital, AP-HP, 75020 Paris, France; 6INSERM U1236, University of Rennes 1, 35000 Rennes, France; 7SITI Laboratory, Rennes University Hospital, 35203 Rennes, France

**Keywords:** hidradenitis suppurativa, verneuil disease, perforator flap, complications, recurrence, propeller flap, island flap, recovery

## Abstract

Introduction: Hidradenitis suppurativa (HS) is a common and debilitating disease, in which the only effective treatment involves a wide excision of the affected skin. Secondary wound healing and skin grafting are two well-known options for managing these defects, but perforator flaps provide a new therapeutic alternative by ensuring reconstructions of large defects, reducing donor site morbidity, and enhancing functional recovery. The aim of this study was to achieve a systematic review of perforator flaps use in HS. Patients and Methods: PubMed and Cochrane databases were searched from 1989 to 2021. The PRISMA statement was used in the study selection process and the review was registered on PROSPERO. Furthermore, patient characteristics, operative technique, complications, and recurrences were searched. Results: Thirty-six articles were selected including 286 patients and 387 flaps. Axillary localization was mostly represented (83.2%). Direct donor site closure was achieved in 99.1% of cases. In total, 15.1% of the flaps presented at least one of the following complications: wound dehiscence (5.5%), partial necrosis (2.9%), hematoma or seroma (2.1%), infection (2.1%), venous congestion (1.8%), and nerve injury (0.3%). Two cases of total necrosis were recorded. Recurrence of the disease was observed in 2.7% of the defects. Conclusions: Pedicled perforator flaps are a reliable and reproducible technique in the reconstruction of HS defects. They are associated with a low recurrence rate while ensuring an effective reconstruction with reduced morbidity and faster recovery compared to the techniques classically used in this indication.

## 1. Introduction

Hidradenitis suppurativa (HS) is a chronic, inflammatory, and recurrent skin disease, whose pathophysiology is still poorly understood [1]. The central pathogenic event is believed to be the occluded hair follicles, provoking an immune response [2] leading to local inflammation of the apocrine gland-bearing areas of the body, most commonly the axillae, inguinal, and anogenital regions. Pro-inflammatory cytokines such as TNF-α are involved [2] and the inflammatory vicious circle finally results in painful subcutaneous nodules and fluctuant draining abscesses with a constant malodorous discharge, impairing dramatically the quality of life [3]. It usually appears after puberty, with a female predominance, and progresses through alternating phases of relapse and remission. It is a frequent but relatively unknown condition, whose prevalence in Europe is estimated to be approximately 1%, resulting in significant healthcare costs [1]. Three stages of increasing severity are described, in order to guide the therapeutic strategy, according to Hurley’s classification [1,4,5]: stage 1, with single or multiple abscess formations, without sinus tracts and scarring; stage 2, associated with single or multiple recurrent abscesses with tract formation and scarring, and widely separated lesions; stage 3 diffuse or near-diffuse involvement, or multiple interconnected tracts and abscesses across the entire area. In all cases, this pathology is significantly associated with smoking and overweight [6]. The interest in HS and the progress in the understanding of its pathophysiology have led to the emergence of several types of medical treatments [7]. Several therapies are now used in current practice from the early stages [8], such as antibiotics (tetracyclines [9,10]), rifampicin/clindamycin association [11,12], ertapenem [10,13]^,^ or retinoids [7,9]. Biologic therapies targeting TNF or other cytokines appear very encouraging especially for more severe stages: so far, adalimumab is the only TNF inhibitor approved by the FDA, but many studies are investigating other molecules [7,9,10]. Theses medical therapies can be used alone or in combination and some authors are attempting to establish algorithms to guide the practice [14]. If the disease is not controlled, it is also interesting to combine them with minimally invasive surgical techniques such as deroofing or limited excisions [1,14,15,16]. However, it has been proven over the last 10 years that in severe stages 3 and some resistant stages 2, wide excision of all the hair-bearing skin is the only treatment associated with a significant reduction in the recurrence rate of the disease [17,18,19,20]. Again, the procedure should be supported by antibiotics to increase the success rate, according to the recommendations of the French Society of Dermatology [21]. These surgical excisions inevitably result in extensive although superficial defects in the joint areas, which must be reconstructed. Several methods of coverage are achievable: direct closure is rarely appropriate given the extent of the defect and correlates with a high rate of recurrence [22,23,24]. Secondary healing, although used in the inguinal region, has several drawbacks: a long healing time especially problematic for the working population [25] requiring complex and costly dressings, and a strong risk of retractile scarring formation limiting joint movements [26]. Skin grafting can theoretically overcome these limitations, but restrictive postoperative immobilization is necessary. Its engraftment rate remains uncertain [27], resulting in side effects similar to those described for secondary healing. Compared to secondary healing, the use of flaps has significantly reduced the duration of hospital stay, the healing time, and the complication rate [28], which nevertheless leads to functional sequelae and a sometimes questionable aesthetic result, especially for muscle flaps. However, their use in HS is growing due to the popularity of perforator flaps, including propeller flaps, which have revolutionized the management of skin defects [29,30,31]. Possibilities are numerous and adaptable to each defect, as demonstrated by the “free-style perforator flap” concept [32]. They provide thin and shapeable tissue while usually allowing direct closure [32,33] and placing scars outside the flexion folds to avoid retractile scarring. Muscle sparing limits the morbidity of harvesting, thereby accelerating postoperative rehabilitation [29,30,34]. To our knowledge, no literature review has yet been performed to identify and synthesize the use of perforator flaps in the covering of defects related to HS. Thus, we performed a systematic review and a meta-analysis toward these ends.

## 2. Patients and Methods

The review was performed in February 2021, guided by the preferred reporting items for systematic reviews and meta-analyses principles [35]. A protocol was published and registered in the international prospective register of systematic reviews National Institute of Health Research (registration: https://www.crd.york.ac.uk/prospero/display_record.php?ID=CRD42020212493).

### 2.1. Eligibility Criteria

Inclusion criteria were all original peer-reviewed studies describing the use of perforator flaps in the reconstruction of defects directly following wide excision in HS, regardless of their location. Exclusion criteria were: articles dealing only with secondary wound resurfacing; defects caused by another etiology than HS; reconstructions not using a perforator flap, such as musculocutaneous flaps; any free flap; duplicate studies; review articles without original data; technical descriptions; letters to the editor without case reports; communications without original data; and studies in a language other than English and French.

PubMed and Cochrane Library electronic databases from 1989 (first description of perforator flap) to 2021 were used. The following keywords were selected: the group ((hidradenitis) OR (hidradenitis suppurativa) OR (hidradenitis suppurative) OR (acne inversa) OR (verneuil) OR (verneuil’s disease)) was associated successively to different types of perforator flaps ((perforator flaps) OR (perforating flaps) OR (propeller flaps) OR (superficial cervical artery perforator flaps) OR (internal mammary artery perforator flaps) OR (thoracoacromial artery perforator flaps) OR (lateral thoracic artery perforator flaps) OR (anterior intercostal artery perforator flaps) OR (lateral intercostal artery perforator flaps) OR (serratus anterior artery perforator flaps) OR (circumflex scapular artery perforator flaps) OR (thoracodorsal artery perforator flaps) OR (dorsal scapular artery perforator flaps) OR (inner arm perforator flaps) OR (posterior arm perforator flaps) OR (deep inferior epigastric artery perforator flaps) OR (superficial inferior epigastric artery perforator flaps) OR (deep circumflex iliac artery perforator flaps) OR (superficial circumflex iliac artery perforator flaps) OR (medial circumflex femoral artery perforator flaps) OR (lateral circumflex femoral artery perforator flaps) OR (anterolateral thigh flaps) OR (internal pudendal artery perforator flaps) OR (external pudendal artery perforator flaps) OR (superior gluteal artery perforator flaps) OR (inferior gluteal artery perforator flaps) OR (parasacral artery perforator flaps)).

### 2.2. Data Collection

Data were collected independently by two researchers (CV, PR). Disagreements were resolved by the senior author (NB). Retrieved information were: author, country, date of publication, type of study, level of evidence, number of patients, age, comorbidities (high blood pressure, smoking, diabetes, body mass index), defect locations and surface, surgical margin, perioperative antibiotic therapy, number, family and type of flaps, surface, level of dissection and degree of rotation of flaps, perforator skeletonization, preoperative perforator detection, rate and type of complications, surgical revision, closure of the donor site, recurrence, and follow-up time. All data were listed in Microsoft Excel 2016 (Microsoft Inc., Redmond, WA, USA).

### 2.3. Assessment of Methodological Quality and Study Outcomes

Each case series and cohort study were assessed for methodological quality using a standardized critical appraisal instrument, the Joanna Briggs Institute (JBI) Critical Appraisal Checklist. Case reports and letters to the editor were not submitted to these appraisal tools. The primary endpoint was to describe the use of perforator flaps in the indication of HS defects in order to assess their safety and reliability. The secondary outcome was to identify risk factors for complications.

All statistical analyses were performed using Prism 5 software (GraphPad Software, San Diego, CA, USA).

Variables are expressed as means ± standard deviations and were compared using Fisher’s exact test. A two-sided *p*-value < 0.05 was considered significant. Subgroup analyses were performed for the predominantly represented axillary location in order to homogenize data and identify risk factors for complications.

## 3. Results

Among 52 identified, 36 [28,36,37,38,39,40,41,42,43,44,45,46,47,48,49,50,51,52,53,54,55,56,57,58,59,60,61,62,63,64,65,66,67,68,69,70] studies were included, as shown in the corresponding flowchart (Figure 1). We included 286 patients with 380 defects and 387 flaps, some defects being reconstructed by two flaps. Most studies had a low level of evidence (Table 1). There were only three prospective studies and no randomized control trial. The methodological quality assessments of the studies are presented in Table 2 (case series) and Table 3 (cohort studies). General patient and flap characteristics are presented in Table 4.

### 3.1. Overview of Techniques

#### 3.1.1. Anatomical Location

The information was found for all flaps (Table 5). Most of them were in the axillary region, with 322 flaps (83.2%); 45 (11.6%) flaps in the anogenital region; 19 (4.9%) in the inguinal region; and 1 (0.3%) in the cervical area.

#### 3.1.2. Defect and Flap Surfaces

Data regarding the defect and flap surfaces were found for 99 and 149 cases, respectively. The mean surface area of the defects was 135.7 ± 82.5 cm^2^ (minimum: 15; maximum: 920). The mean surface area of the flaps was 128.3 ± 70.1 cm^2^ (minimum: 20; maximum: 374). This difference in the extreme values is related to the fact that some articles specified only the surface area of the defect and not those of the flaps used and vice versa.

#### 3.1.3. Flap Types

Data are presented in Table 6. The most frequently used flap type was the propeller perforator flaps in 224 cases (62.9%). We also found 43 advancement flaps (12.1%), 38 keystones (10.7%), 36 peninsular flaps [71,72] (10.1%), 14 VY flaps (3.9%), and 1 interpolation flap (0.3%). Data were collected for 324 flaps (83.7%). The two most commonly used flaps were TDAP and PAP in 48,5% (n = 157) and 21,3% (n = 69) of cases, respectively. We also identified 24 IAPs (7.4%), 10 CSAPs (3.1%), 9 LTAPs (2.8%), 7 LICAPs (2.2%), 3 SAAPs (0.9%), and 1 DSAP (0.3%), which were applied to axillary and cervical defects, as well as 17 SGAP (5.2%), 12 IGAP (3.7%), 4 PFAP (1.2%), 4 SIEAP (1.2%), 4 ALT (1.2%), and 3 MCFAP (0.9%) for inguinal and anogenital reconstruction.

#### 3.1.4. Flap Procurement and Donor Site Closure

Preoperative perforator detection was mentioned in 26 articles (238 flaps). Doppler mapping was performed in 197 cases (82.8%). Perforator skeletonization only referred to propeller flaps, advancement flaps, peninsular flaps, and interpolation flaps. It was found in all of them, and skeletonization was performed in 100% of cases until satisfactory mobilization of the flap was obtained. Flap harvesting was specified for 206 flaps (propeller flaps, advancement flaps, peninsular flaps, and interpolation flaps), i.e., 61.5%. For the flaps involved, the dissection was subfascial in 172 cases (83.5%) and suprafascial in 34 cases (16.5%). For donor site closure, data were found for 322 flaps (83.2%). In total, 319 donor sites were managed with primary closure (99.1%). Three donor sites required coverage using skin grafting (0.9%).

#### 3.1.5. Complications and Revision Surgery

Complication data were identified for 383 flaps (99%) and are presented in Table 7. In total, 58 complications were reported: 21 wound dehiscence (5.5%), 11 partial flap necroses (2.9%), 2 total flap necroses (0.5%), 8 infections (2.1%), 7 venous congestions (1.8%), 5 hematomas (1.3%), 3 seromas (0.8%), and 1 nerve injury (0.3%). Revision surgery was specified in 31 articles (302 flaps). It was required for 11 flaps (3.6% of cases). Regarding recurrence and follow-up duration, information was collected for 336 defects (88.4%). A recurrence rate of 2.7% was observed (n = 9). Follow-up time was specified for 291 flaps, with a mean of 16.6 ± 9.9 months (minimum: 2; maximum: 60).

### 3.2. Risk Factors Analysis for Axillary Location

#### 3.2.1. Based on the Flap Type

The complication rate based on axillary location is presented in Table 8. The information was found for 83.9% of the axillary location flaps (n = 270). No statistical difference was demonstrated (Figure 2). The use of propeller flaps was not associated with a significant over-risk of complication compared to peninsular flaps (*p* = 0.0851), VY flaps (*p* = 0.2135), advancement flaps (*p* = 0.4910), and keystones (*p* = 0.5346).

#### 3.2.2. Based on the Type of Perforator

The complication rate according to the type of perforator for the axillary location is presented in Table 9. This data was specified for 80.1% of the axillary flaps (n = 258). No significant difference was found depending on the type of perforator (Figure 3): TDAP flaps were no more complicated than PAP flaps (=0.3201), CSAP flaps (*p* = 0.3665), LTAP flaps (*p* = 0.5865), or IAP flaps (*p* > 0.9999).

#### 3.2.3. Based on the Type of Perforator among the Propeller Flaps

The complication rate based on the type of perforator among the axillary propeller flaps is presented in Table 10. The information was found for 81% of the axillary propeller flaps (n = 158). No significant difference was shown according to the type of perforator in this subpopulation (Figure 4). The complication rate found for TDAP flaps was not significantly higher than for CSAP flaps (=0.3586), PAP flaps (*p* = 0.5935), IAP flaps (*p* > 0.9999), or LTAP flaps (*p* > 0.9999).

## 4. Discussion

HS-related defects represent a true challenge for the plastic surgeon, as they require a large resurfacing with subsequent additional morbidity to the original disease. Conventional techniques represented by secondary healing and skin grafting have several limits with a slow recovery time which impacts the resumption of professional activities. Our review showed an overall recurrence rate of 2.7% after wide excision of the lesions and coverage with a perforator flap, for a mean follow-up time of 16.6 months. Secondary healing has the advantage of not increasing the operative time, but is associated with problematic skin retraction in the peri-articular regions [26], as well as a considerable healing time, up 2–3 months [24,25]: in the active population of patients, this represents a delayed return to work and significant healthcare costs. Moreover, the recurrence rates observed in the literature after secondary wound healing are not negligible: Ovadja et al. [23] observed an 11% recurrence rate; Deckers et al. [73] reported a recurrence rate of 37.6% in a total of 253 cases. The median time to recurrence in this last retrospective study was 6 months (IQR: 3.0–13.0). The use of skin grafts after wide HS excision is also widely adopted: nevertheless, the uncertainty of the graft take can lead to a period of secondary healing [25], immobilization is necessary to promote success [74], and there is non-negligible morbidity of the donor site. Moreover, several studies report a failure rate of over 40% in this indication [27,28,75], and the time to return to work activity was significantly longer than after reconstruction with a perforator flap [28]. Regarding recurrence after skin grafting, the meta-analyses of Ovadja et al. [23] and Mehdizadeh et al. [20] found rates of 2% and 6%, respectively, even up to 20% in some studies [24,76]. In our review, with the use of perforator flaps, we found an overall complication rate of 15.1%, i.e., 58 complications for 383 flaps, with only 2 total necroses. In total, 12 cases of partial necrosis were observed, the most frequent complication being wound dehiscence. These complications led to revision surgery in 3.6% of cases. When comparing the use of perforator flaps in HS to other indications, a recent literature review reported an overall complication rate of 12.3% for 432 thoracic pedicled perforator flaps used for reconstruction after breast cancer [77]. In the axillary region, Jiang et al. [78] described a complication rate of 12.5% in a series of 32 TDAPs used to treat axillary scar contractures. In a retrospective study, Brunetti et al. [79] reported a complication rate of 23.1% in a total of 130 consecutive flaps for reconstructions of various locations. The majority of these defects were related to carcinologic resection (117 cases). Finally, in the lower limbs, the complication rate found after coverage of burns and traumatic, tumoral, or infectious lesions with perforator flaps was 23% [80]. In the axillary region, our review of 183 perforator propeller flaps showed a complication rate of 20.2%, with 37 complications. A meta-analysis by Florczak et al. [81] showed an average complication rate of 9.9% for 182 flaps in the thoracic region. In the review by Lazzeri et al. [82] on 288 propeller perforator flaps in the head and neck, trunk, and upper limb regions, the complication rate was 13.8%. Complications were more frequent when the defects were located at the extremities: regarding the upper limb, 23.9% of the flaps were complicated in the review by Vitse et al. [83] (on 117 defects): for the lower limb, 25.2% of flap complications were reported in the meta-analysis by Bekara et al. [84], on a total of 428 propeller flaps. The origin of these complications was mainly traumatic. The complication rate of perforator flaps in HS thus appears to be comparable to the complication rates of the same type of flaps encountered for other surgical indications and locations. In this review, we analyzed the complication rates for the different families and types of perforator flaps in the axilla. Similarly, the type of perforator flap did not appear to be a risk factor for complication, both in the total population of axillary reconstructions and in the subgroup of propeller flaps.

The surgical procedure is therefore not trivial, and the complication rate is significant, especially in case of highly active disease. An interesting approach is the combination of a surgical procedure with an adjuvant or even neo-adjuvant treatment. For moderate to severe HS stages, immunomodulatory drugs seem to have certain efficiency. If Adalimumab is currently the only validated biotherapy, Janus kinase and C5a inhibitors and antagonists are under investigation [85].

The SHARPS randomized clinical trial [86] showed an improved clinical response with adalimumab vs. placebo perioperatively for all treated body regions. However, the results were poorly significant (*p*-value of 0.049), the follow-up period was relatively short (12 weeks), and the trial highlighted the potential complications of this biotherapy. Further studies should assess better the effects of combined treatment with surgery and adalimumab or future approved molecules. A phase IV trial is currently underway that may provide guidance on the preoperative use of adalimumab to improve local status at the time of surgery [87].

We recognize several limitations to our study. First of all, as this was a systematic review, we had to include articles with a low level of scientific evidence given that they were based on retrospective studies. Our team is currently involved in a randomized multicentric trial comparing the outcomes of secondary wound healing versus pedicled perforator flaps after radical HS excision [88] to bring better evidence in this field. Secondly, the complication rate may have been underestimated because of a subjective assessment depending on the surgeon. The lack and heterogeneity of data concerning surgical techniques were recurrent problems, which may have compromised the identification of risk factors for complications according to the flap’s characteristics. As our review was observational, a randomized controlled trial with standardized data collection would be needed to establish a management decision algorithm based on risk factors for complications.

In conclusion, our study highlights both the safety and reliability of perforator flaps for covering defects following the excision of HS. Perforator flaps offer a suitable option to cover wide defects after HS surgical treatment. It is a method that ensures a good quality reconstruction, is custom-made, reliable, and reproducible, as has already been demonstrated for various etiologies. Its low morbidity, due to the respect for the underlying muscle and the direct closure of the donor site, makes it a technique of choice for the coverage of these defects in patients whose quality of life is already significantly impacted by the disease.

## Figures and Tables

**Figure 1 jcm-11-05813-f001:**
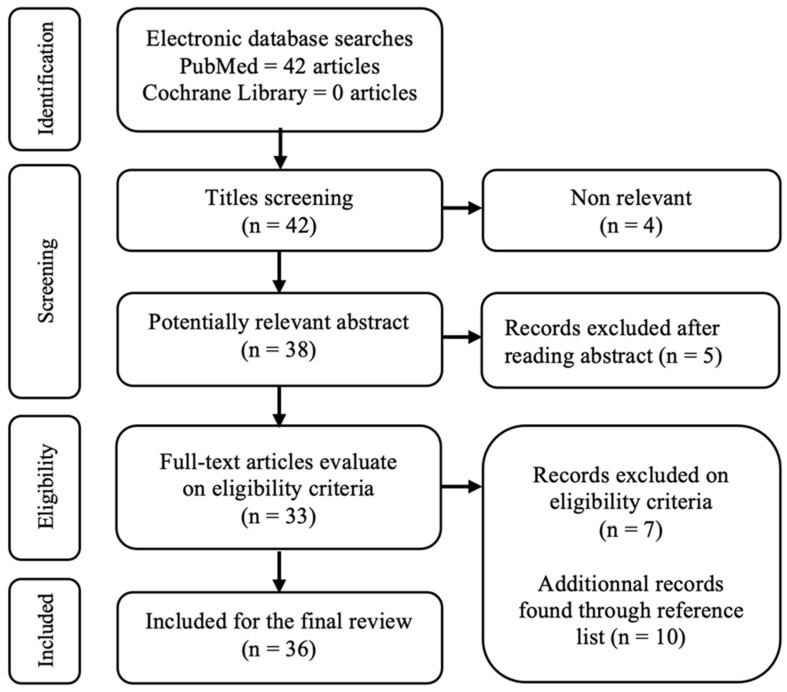
PRISMA flowchart or the meta-analysis.

**Figure 2 jcm-11-05813-f002:**
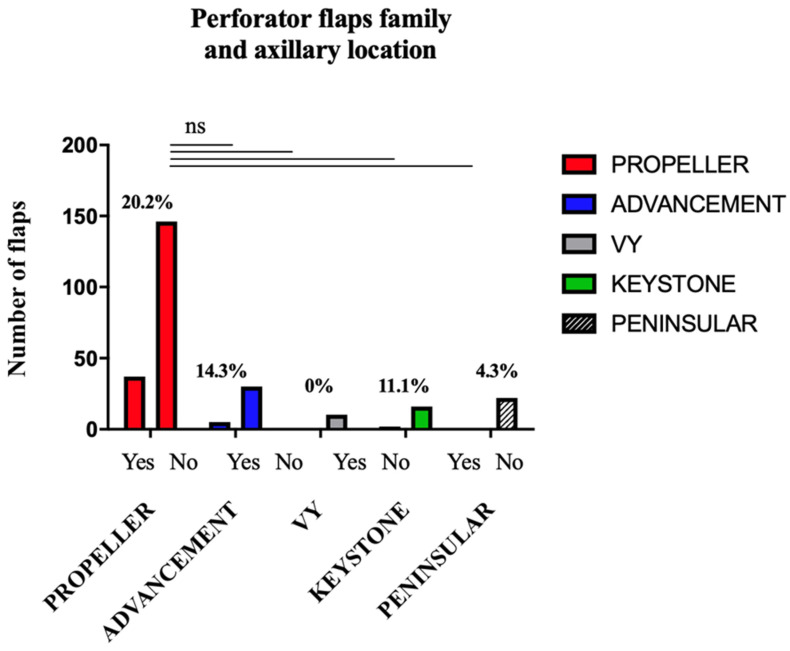
Complication Rate based on the Flap Family for Axillary Location.

**Figure 3 jcm-11-05813-f003:**
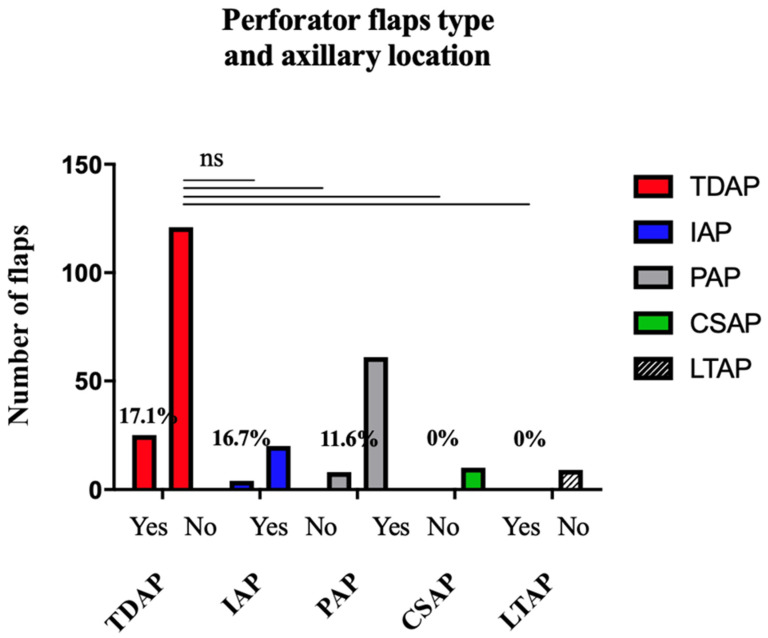
Complication Rate based on the Flap Type for Axillary Location.

**Figure 4 jcm-11-05813-f004:**
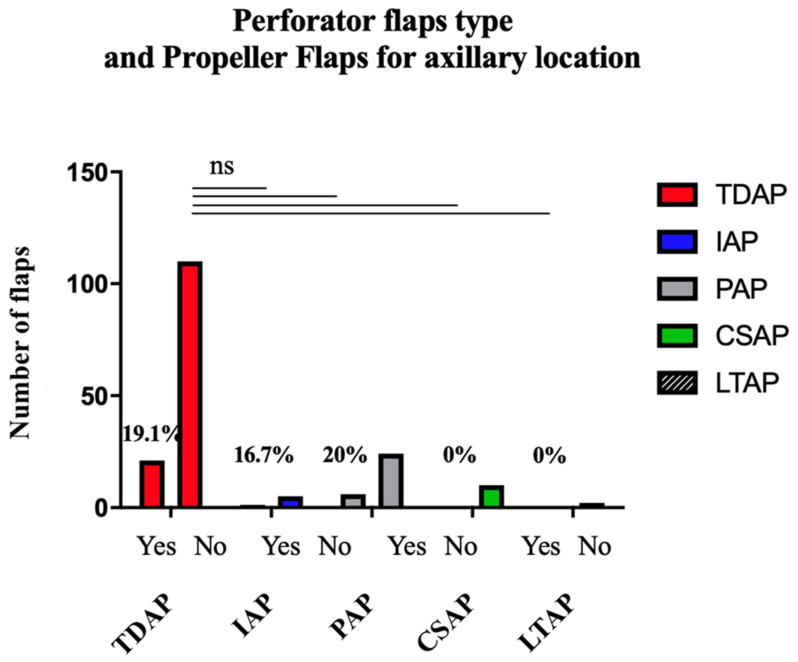
Complication Rate based on the Flap Type among the Propeller Flaps for Axillary Location.

**Table 1 jcm-11-05813-t001:** Presentation of Included Studies, with Level of Evidence.

Study	Country	Study Design	EBM	Number of Patients
Elliot et al., 1992 [36]	United Kingdom	Case series	4	17
Amarante et al., 1996 [37]	Portugal	Case series	4	6
Schwabegger et al., 2000 [38]	Austria	Case series	4	6
Geh et al., 2002 [39]	United Kingdom	Case series	4	4
Guerra et al., 2004 [40]	United States	Case series	4	2
Rehman et al., 2005 [41]	United Kingdom	Case series	4	3
Sharma et al., 2006 [42]	India	Case series	4	6
Rees et al., 2007 [43]	United Kingdom	Case report	5	1
Dabernig et al., 2007 [44]	United Kingdom	Case series	4	2
Laredo Ortiz et al., 2007 [45]	Spain	Case series	4	2
Ayhan et al., 2008 [46]	Turkey	Case series	4	3
Kishi et al., 2009 [47]	Japan	Case series	4	4
Laredo Ortiz et al., 2010 [48]	Spain	Case series	4	16
Unal et al., 2011 [49]	Turkey	Case series	4	12
Busnardo et al., 2011 [50]	Brazil	Prospective cohort	5	12
Sever et al., 2012 [51]	Turkey	Case series	4	2
Hallock 2013 [52]	United States	Case series	4	2
Egemen et al., 2013 [53]	Turkey	Case series	4	11
Alharbi et al., 2014 [54]	France	Case series	4	10
Wormald et al., 2014 [28]	United Kingdom	Prospective cohort	2	15
Mehrotra 2015 [55]	India	Letter to the editor	4	NR
Baghaki et al., 2015 [56]	Turkey	Case report	5	1
Schmidt et al., 2015 [57]	Austria	Case series	4	20
Haq et al., 2015 [58]	Pakistan	Case report	5	1
Hoang et al., 2016 [59]	United States	Case report	5	1
Ching et al., 2017 [60]	Australia	Case series	4	4
Nail-Barthelemy et al., 2019 [61]	France	Case series	4	13
Lakshmana Rao et al., 2018 [62]	India	Case series	4	8
Elgohary et al., 2018 [63]	Egypt	Prospective cohort	3	20
Marchesi et al., 2018 [64]	Italy	Case series	4	12
Elboraey et al., 2019 [65]	Kuwait	Case series	4	6
Sirvan et al., 2019 [66]	Turkey	Case series	4	14
Rodriguez et al., 2019 [67]	Colombia	Case series	4	2
Kim et al., 2020 [68]	Korea	Case report	5	1
Virág et al., 2020 [69]	Romania	Case series	4	21
Marchesi et al., 2021 [70]	Italy	Case series	4	26

**Table 2 jcm-11-05813-t002:** Assessment of Methodological Quality of Case Series by the JBI Critical Appraisal Tool.

Case Series	Were There Clear Criteria for Inclusion in the Case Series?	Was the Condition Measured in a Standard, Reliable Way for All Participants Included in the Case Series?	Were Valid Methods Used for Identification of the Condition for All Participants Included in the Case Series?	Did the Case Series Have Consecutive Inclusion of Participants?	Did the Case Series Have Complete Inclusion of Participants?	Was There Clear Reporting of the Demographics of the Participants in the Study?	Was There Clear Reporting of Clinical Information of the Participants?	Were the Outcomes or Follow-Up Results of Cases Clearly Reported?	Was There Clear Reporting of the Presenting Site(s)/Clinic(s) Demographic Information?	Was Statistical Analysis Appropriate?
Elliot et al., 1992 [36]	no	NA	no	no	no	yes	no	yes	no	NA
Amarante et al., 1996 [37]	no	NA	no	no	no	yes	no	yes	no	NA
Schwabegger et al., 2000 [38]	no	no	no	yes	yes	no	no	yes	no	NA
Geh et al., 2002 [39]	yes	NA	yes	no	no	yes	no	yes	yes	NA
Guerra et al., 2004 [40]	yes	NA	no	yes	yes	no	no	no	no	NA
Rehman et al., 2005 [41]	no	NA	no	no	no	yes	no	yes	no	NA
Sharma et al., 2006 [42]	yes	NA	yes	yes	yes	yes	no	yes	no	NA
Rees et al., 2007 [43]	no	no	no	no	no	yes	no	no	no	NA
Laredo Ortiz et al., 2007 [45]	yes	NA	no	yes	yes	yes	no	no	no	NA
Ayhan et al., 2008 [46]	yes	NA	no	yes	yes	yes	no	no	no	NA
Kishi et al., 2009 [47]	yes	NA	yes	no	no	yes	no	yes	no	NA
Laredo Ortiz et al., 2010 [48]	yes	NA	yes	no	no	yes	no	yes	no	NA
Unal et al., 2011 [49]	yes	NA	yes	yes	yes	yes	no	yes	no	NA
Sever et al., 2012 [51]	yes	NA	yes	yes	yes	yes	no	no	no	NA
Hallock 2013 [52]	yes	NA	yes	yes	yes	yes	no	yes	no	NA
Egemen et al., 2013 [53]	yes	NA	yes	no	no	yes	no	yes	no	NA
Alharbi et al., 2014 [54]	yes	NA	yes	yes	yes	yes	no	yes	yes	NA
Mehrotra 2015 [55]	no	NA	no	no	no	no	no	no	no	NA
Schmidt et al., 2015 [57]	yes	NA	no	yes	yes	yes	no	yes	no	NA
Ching et al., 2017 [60]	yes	NA	yes	yes	yes	yes	no	yes	no	NA
Nail-Barthelemy et al., 2019 [61]	yes	NA	yes	yes	yes	yes	yes	yes	yes	NA
Lakshmana Rao et al., 2018 [62]	yes	NA	yes	no	no	yes	no	yes	no	NA
Marchesi et al., 2018 [64]	yes	NA	yes	yes	yes	yes	no	yes	no	NA
Elboraey et al., 2019 [65]	yes	NA	yes	yes	yes	yes	yes	yes	yes	NA
Sirvan et al., 2019 [66]	yes	NA	yes	no	no	yes	no	yes	yes	NA
Rodriguez et al., 2019 [67]	yes	NA	yes	no	no	yes	no	yes	no	NA
Virág et al., 2020 [69]	yes	NA	yes	yes	yes	yes	no	yes	no	NA
Marchesi et al., 2021 [70]	yes	NA	yes	yes	yes	yes	no	yes	no	NA

NA = not applicable.

**Table 3 jcm-11-05813-t003:** Assessment of Methodological Quality of Cohort Studies by the JBI Critical Appraisal Tool.

Cohort Studies	Were the Two Groups Similar and Recruited from the Same Population?	Were the Exposures Measured Similarly to Assign People to both Exposed and Unexposed Groups?	Was the Exposure Measured in a Valid and Reliable Way?	Were Found Confounding Factors Identified?	Were Strategies to Deal with Confounding Factors Stated?	Were the Groups/Participants Free of the Outcome at the Start of the Study (or at the Moment of Exposure)?	Were the Outcomes Measured in a Valid and Reliable Way?	Was the Follow up Time Reported and Sufficient to Be Long Enough for Outcomes to Occur?	Was Follow up Complete, and If Not, Were the Reasons to Loss Follow up Described and Explored?	Were Strategies to Address Incomplete Follow up Utilized?	Was Appropriate Statistical Analysis Used?
Busnardo et al., 2011 [50]	yes	yes	yes	no	NA	no	yes	yes	no	NA	yes
Wormald et al., 2014 [28]	yes	NA	NA	no	NA	no	yes	yes	Unclear	Unclear	yes
Elgohary et al., 2018 [63]	no	yes	yes	Unclear	Unclear	no	yes	yes	NA	NA	yes

NA = not applicable.

**Table 4 jcm-11-05813-t004:** Patients and Operative Characteristics.

	Value	%
Articles	36	
Patients	286	
Flaps	387	
Age (years)		
Articles including data	32	88.9% (1)
Mean +/− 95% CI	35.6 +/− 0.91	
Range	16–76	
Female sex		
Number of patients	125	48.3% (1)
Articles including data	31	86.1% (1)
Smoking		
Number of patients	66	55% (1)
Articles including data	24	66.7% (1)
HBP		
Number of patients	5	7% (1)
Articles including data	8	22.2% (1)
Diabetes		
Number of patients	16	12.6% (1)
Articles including data	12	33.3% (1)
BMI (kg/m^2^)		
Articles including data	6	16.7% (1)
Mean +/− 95% CI	27.9 +/− 1.20	
Range	18.17–45.7	
Overweight/obesity		
Number of patients	63	67.7% (1)
Articles including data	9	25% (1)
Location		
Articles including data	36	100% (2)
Axillary	322	83.2% (2)
Inguinal	19	4.9% (2)
Anogenital	45	11.6% (2)
Cervical	1	0.3% (2)
Hurley stage		
Articles including data	12	33.3% (2)
Mean +/− 95% CI	2.8 +/− 0.0	
Range	2–3	
Lateral surgical margin		
Articles including data	32	88.9% (2)
Affected skin	142	37.5% (2)
Surgical margin around affected skin	77	20.3% (2)
Hairy skin	157	41.4% (2)
Surgical margin around hairy skin	3	0.8% (2)
Deep surgical margin		
Articles including data	31	86.1% (2)
Affected area	243	66.6% (2)
Subcutaneous tissue	5	1.4% (2)
Subcutaneous tissue excluding fascia	54	14.8% (2)
Subcutaneous tissue including fascia	44	12.1% (2)
Surgical margin around affected area	19	5.2% (2)
Peri operative antibiotic therapy		
Number of flaps	126	100% (2)
Articles including data	9	25% (2)
Surface of flap (cm^2^)		
Articles including data	15	41.7% (2)
Mean +/− 95% CI	128.3 +/− 11.3	
Range	20–374	

**Table 5 jcm-11-05813-t005:** Locations of Flaps.

Location	Flaps (n)	Flaps (%)
Axillary	322	83.2
Anogenital	45	11.6
Inguinal	19	4.9
Cervical	1	0.3
Total	387	100

**Table 6 jcm-11-05813-t006:** Types of Flaps.

Location	Flaps (n)	Flaps (%)
Total	356	100
Propeller	224	62.9
Advancement	43	12.1
Peninsular	36	10.1
Interpolation	1	0.3
VY	14	3.9
Keystone	38	10.7

**Table 7 jcm-11-05813-t007:** Complications.

Location	Flaps (n)	Flaps (%)
Total	58	15.1
Total necrosis	2	0.5
Partial necrosis	11	2.9
venous congestion	7	1.8
Wound deshiscence	21	5.5
Hematoma	5	1.3
Seroma	3	0.8
Infection	8	2.1
Nerve Injury	1	0.3

**Table 8 jcm-11-05813-t008:** Complication rate based on the flap type for axillary location.

Type of Flap	Flaps (n)	Complications (n)	Complications (%)
Total	322	48	14.9
Propeller	183	37	20.2
Advancement	35	5	14.3
Peninsular	23	1	4.3
Interpolation	1	0	0
Keystone	18	2	11.1
VY	10	0	0
NR	52	3	5.8

**Table 9 jcm-11-05813-t009:** Complication rate based on the perforator type for axillary location.

Type of Perforator	Flaps (n)	Complications (n)	Complications (%)
Total	322	48	14.9
TDAP	146	25	17.1
PAP	69	8	11.6
IAP	24	4	16.7
CSAP	10	0	0
LTAP	9	0	0
SAAP	0	0	0
LICAP	0	0	0
NR	64	11	17.2

TDAP, Thoraco Dorsal Artery Perforator; PAP, Posterior Arm Perforator; IAP, Inner Arm Perforator; CSAP, Circumflex Scapular Artery Perforator; LTAP, Lateral Thoracic Artery Perforator; SAAP, Serratus Anterior Artery Perforator; LICAP, Lateral Artery Perfrator.

**Table 10 jcm-11-05813-t010:** Complication rate based on the perforator type among the propeller flaps for axillary location.

Type of Perforator	Flaps (n)	Complications (n)	Complications (%)
Total	195	39	20
TDAP	110	21	19.1
PAP	30	6	20.0
IAP	6	1	16.7
CSAP	10	0	0
LTAP	2	0	0
SAAP	0	0	0
LICAP	0	0	0
NR	37	11	29.7

TDAP, Thoraco Dorsal Artery Perforator; PAP, Posterior Arm Perforator; IAP, Inner Arm Perforator; CSAP, Circumflex Scapular Artery Perforator; LTAP, Lateral Thoracic Artery Perforator; SAAP, Serratus Anterior Artery Perforator; LICAP, Lateral Artery Perfrator.

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
