# Peer review of "Outcomes and Reliability of Perforator Flaps in the Reconstruction of Hidradenitis Suppurativa Defects: A Systemic Review and Meta-Analysis"

_jcm, 2022, doi:10.3390/jcm11195813_

Round 1
Reviewer 1 Report
The work is nice, very well performed and manuscript is very well written.
1. Minor grammar and spelling check and corrections required.
2. Include more related latest references
Author Response
Thank you very much for your comments and piece of advice.
- As suggested, the article has been edited by a native English speaker, for grammatical error correction. Some typos have been corrected as well.
- We updated the bibliography in the introduction by adding several references related to alternative treatments.
Reviewer 2 Report
I find it an interesting job and well organised. It is well presented and the Enlish language is adequate. Please correct in line 315 "well", instead of "lwell". Please consider adding in line 54 ("this pathology is significantly associated with smoking...) also the reference:
"Thyroid disease and active smoking may be associated with more severe hidradenitis suppurativa: Data from a prosepective, cross- sectional, single-center study.
Aikaterini I. Liakou, Georgios Kontochristopoulos et al."
Author Response
Thank you for your comments and meticulous review. We corrected the typos and the grammar as suggested. We added the reference from Aikaterini et al. in the introduction as kindly recommended.
Reviewer 3 Report
Vaillant et al. performed a systemic analysis regarding reconstruction options for hidradenitis suppurativa (HS).
Using biologics such as TNF-alpha inhibitors holds promise for HS treatment. However, for advanced lesions with fibrosis and fistulas, there have been no medications that could bring about complete remission (CR). Thus, evaluating the legitimacy of surgical procedures aiming at CR is essential.
This manuscript is generally well-written, straightforward, and supports the use of perforator flaps for reconstructing the defect following HS excision.
However, the major drawback of this study is in that they ended up collecting pieces of anecdotal evidence, pausing the question if the content could provide novelty, even if the authors claimed ‘a new therapeutic alternative’, to the research field.
The reviewer raises some concerns.
Major concerns.
1. Background information requires update: in the introduction section, the authors should include information regarding non-invasive HS treatment measures, such as biologics or other anti-inflammatory drugs. They need to refer to recent reviews from the dermatology field, as in ref. 1. Because complete resection can be a last resort for HS treatment in the near future, it is also important for the authors to review the adjuvant therapy using biologics.
2. The reviewer thinks that the surgical field with extensive inflammation generally confers poor outcomes, and the flap technique tends to be protected from surgical site infection compared with skin grafting or primary closure. Alternatively, delayed grafting may be recommended when performing skin grafting. The authors should state explicitly from the evidence available that flaps are superior to skin grafts.
3. The authors focused on procedural aspects, but surgical site information can be crucial for optimal outcomes. Authors should add information regarding HS clinical grades other than Hurley (Sartorius, or IHS4) or serum CRP levels in connection with clinical outcomes, if available.
Minor concerns.
1. L40 whose, not which
2. L41 occluded, not occlusion
3. L127 series, not serie
4. L315 well, not lwell
Author Response
Thank you very much for your very constructive review and your questions. We have tried to answer each of your pertinent remarks.
Please find the answers to the concerns you highlighted:
Major concerns :
- As suggested, we have reworked the introduction section and added several references regarding non-invasive treatment and combined treatment, including biologics and other anti-inflammatory drugs.
- We strongly agree with the reviewer that flap reconstruction confers better protection from surgical site infection in comparison with primary closure or primary skin grafting. The available evidence comparing flaps to secondary guided wound healing and skin grafting is very poor. We are currently involved in a randomized multicentric study comparing perforator flap surgery vs. conventional treatment after extended HS excision, to better compare the outcomes of both approaches. We added this information to the conclusion, but we believe that for now, the level of evidence is weak even if the results seem obviously in favor of the flap reconstructive technique.
- We agree with the reviewer that the patients should be considered globally and that surgeons should not focus only on the surgical site. Unfortunately, the articles included in this review did not give information relative to systemic inflammation (CRP, Blood sedimentation rate…) not or other clinical scales. This should be evaluated pre-operatively.
Minor concerns :
We fixed the grammar errors and typos and the manuscript has been edited by a native English speaker. Thank you.
Reviewer 4 Report
Dear Authors,
This article is interesting. It will contribute to the literature. I have some suggestions to make it a more targeted and nice article;
1- The language of the article needs to be improved (there are grammatical errors and typos). It is recommended that the article is edited by a native English speaker.
2- In the introduction and discussion, I suggest that you mention the combination of biologic agent treatments (such as anti-TNF agents) and surgery, which have had very successful results in the treatment of HS in recent years.
2- The discussion is very shallow, I am waiting for a wide discussion part with combination treatments (medical and surgical).
Best wishes...
Author Response
Thank you very much for your comments and suggestions.
Please find our answers to each of your remarks :
- The language has been edited, and the grammar and typos have been fixed by a native English speaker.
- We have addressed the subject of biologic agent treatments alone and combined with surgery in the introduction and discussion, and specified that the current evidence in stade 3 and resistant stade 2 patients are broadly in favor of wide excision of the hair-bearing skin for long-term improvement.
- We added a paragraph discussing the combination treatments with surgery and biotherapies as suggested. Indeed, since the best medical treatment results so far for moderate to severe HS have been shown by using immunomodulatory drugs, we focused on the available evidence of this combination.
We hope that our manuscript will gain your acceptance.
Sincerely
Round 2
Reviewer 3 Report
Authors have addressed the issues raised by this reviewer appropriately.